# Structural Characterization of the Milled-Wood Lignin Isolated from Sweet Orange Tree (*Citrus sinensis*) Pruning Residue

**DOI:** 10.3390/polym15081840

**Published:** 2023-04-11

**Authors:** Mario J. Rosado, Jorge Rencoret, Ana Gutiérrez, José C. del Río

**Affiliations:** Instituto de Recursos Naturales y Agrobiología de Sevilla, CSIC, Avda. Reina Mercedes, 10, 41012 Seville, Spain

**Keywords:** orange tree pruning, lignocellulose, milled-wood lignin, 2D-NMR, pyrolysis

## Abstract

The pruning of sweet orange trees (*Citrus sinensis*) generates large amounts of lignocellulosic residue. Orange tree pruning (OTP) residue presents a significant lignin content (21.2%). However, there are no previous studies describing the structure of the native lignin in OTPs. In the present work, the “milled-wood lignin” (MWL) was extracted from OTPs and examined in detail via gel permeation chromatography (GPC), pyrolysis-gas chromatography/mass spectrometry (Py-GC/MS), and two-dimensional nuclear magnetic resonance (2D-NMR). The results indicated that the OTP-MWL was mainly composed of guaiacyl (G) units, followed by syringyl (S) units and minor amounts of *p*-hydroxyphenyl (H) units (H:G:S composition of 1:62:37). The predominance of G-units had a strong influence on the abundance of the different linkages; therefore, although the most abundant linkages were β–*O*–4′ alkyl–aryl ethers (70% of total lignin linkages), the lignin also contained significant amounts of phenylcoumarans (15%) and resinols (9%), as well as other condensed linkages such as dibenzodioxocins (3%) and spirodienones (3%). The significant content of condensed linkages will make this lignocellulosic residue more recalcitrant to delignification than other hardwoods with lower content of these linkages.

## 1. Introduction

Sweet orange (*Citrus sinensis* L. Osbeck) is a tree from the Rutaceae family that is widely cultivated worldwide for its fruit, with a total cultivated area of 3.93 M ha and a global production of 75.56 M tonnes in 2021 [1]. The world’s largest orange producers are Brazil (16.5 M tonnes), India (10.2 M tonnes), and China (7.5 M tonnes), followed by Mexico (4.6 M tonnes), the United States (4.3 M tonnes), Egypt (3.5 M tonnes), and Spain (3.3 M tonnes) [1].

Tree pruning is an important process in the management of fruit trees, including orange tree plantations. The annual pruning of mature trees is necessary to remove dead plant matter, prevent aging, allow light and air to enter through the canopy, control the growth of new shoots, and increase the fruit productivity [2,3]. Considering the enormous world production of oranges, pruning results in huge amounts of orange tree pruning (OTP) residues being produced annually. For example, in the case of the Valencia Late variety, they can produce an average of 4.7 tonnes ha^−1^ of dry biomass, depending on the cultivar and environmental factors [4], which, if extrapolated to the global cultivated area, will represent more than 18.4 M tonnes of OTP residues per year being produced worldwide. OTP residue, along with other similar pruning residues derived from other fruit trees, is usually burned in landfills to incorporate the ashes into the soil as amendments, although this agricultural practice is generally prohibited by local authorities for environmental reasons [3]. This residue is a lignocellulosic material that has also been used for a variety of purposes, such as the production of biofuels and bioproducts, as a soil amendment, for the formation of pellets, and as sustainable reinforcement fibre for biopolyethylene, as well as for the manufacturing of pulp and paper [5,6,7,8,9,10]. However, due to their wide availability and low price, as well as their high lignin and carbohydrate contents, OTPs can be considered excellent candidates for use as feedstocks to produce biofuels and biomaterials in the context of the so-called lignocellulosic biorefineries [7,11,12,13,14,15].

However, and despite the high amounts of OTPs available worldwide and their varieties of uses, there are no previous studies addressing the structure of the native lignin in this lignocellulosic material. Only one previous work was published that reported the structural characteristics of the lignin obtained from the black liquors produced after soda/anthraquinone (soda/AQ) cooking of OTPs [11]. However, it is evident that the structure of the lignin in the black liquors was greatly modified during the cooking process and was no longer representative of the native lignin, so the structure of the native lignin of OTPs is still unknown. The complete and efficient utilization of any lignocellulosic biomass requires the detailed knowledge of its components, in particular, the structure of the lignin polymer, which has a negative impact during industrial processing in cellulose-based industries (such as obtaining cellulose for pulp and paper or bioethanol production). Due to its high recalcitrance, the presence of lignin is a limiting factor in the production of cellulose, and pretreatment methods are necessary for removing or softening the lignin in order to access the carbohydrates [16,17]. The efficiency of any pretreatment method depends to a large extent on the structure of the lignin polymer; therefore, the knowledge of its structure is necessary to be able to develop suitable pretreatment methods that allow for the efficient removal and/or modification of the lignin. In addition, the lignin polymer itself has inherent value and can be used in a biorefinery, as recognized by the rise of “lignin-first” approaches, which seek to preserve the lignin for further processing into different products and commodities [13,14,15,18,19]. Therefore, understanding the structure of the lignin of OTP residue is imperative for maximizing the valorisation and the different uses of this interesting lignocellulosic waste material.

In this context, the main objective of this study is, therefore, to accomplish the detailed structural characterisation of the native lignin in OTP residue. To avoid and minimize the potential chemical modifications produced in the lignin structure during the isolation process, the lignin was isolated with aqueous dioxane according to classical methods [20], which produces a lignin preparation (so-called “milled-wood lignin”, MWL) that is essentially chemically unaltered and is considered to reflect the native plant’s lignin [21]. The OTP-MWL was then analysed using different analytical techniques, including gel permeation chromatography (GPC), pyrolysis-gas chromatography/mass spectrometry (Py-GC/MS), and two-dimensional nuclear magnetic resonance (2D-NMR). The information obtained from this study is expected to bolster the utilization of this low-cost and readily available waste material as a viable feedstock in future lignocellulosic biorefineries.

## 2. Materials and Methods

### 2.1. Samples and Analysis of the Main Constituents 

The orange (*C. sinensis* L. Osbeck) tree pruning (OTP) residue studied in this work consisted mainly of small, thin branches of around 3–5 cm in diameter, which were collected in July 2021 from a plantation in Seville (Spain). OTPs were manually chipped and homogenized and then air-dried. Finally, they were milled in an IKA MF 10 knife mill (IKA, Staufen, Germany), successively passing through 2 mm and 1 mm sieves. The content of the OTP extractives was determined by consecutive Soxhlet extraction with acetone (8 h), methanol (8 h), and water (4 h) and gravimetric determination of the extracts after evaporation in a rotary evaporator. The lignin content was measured according to the Klason method from the resulting solid residue after hydrolysis with sulfuric acid from previously extracted material, according to Tappi method T222 om-88 [22]. The Klason lignin content was subsequently corrected for protein and ash. The acid-soluble lignin content was measured according to Tappi method UM 250 using a spectrophotometer at 205 nm and an extinction coefficient of 1100 cm^−1^ g^−1^ [22]. The content of holocellulose (which included both the hemicelluloses and cellulose) was determined from the extractive-free sample via delignification over a period of 4 h, using the acid chlorite method [23]. The content of cellulose was measured after elimination of the hemicelluloses from the holocellulose by an extraction with alkali [23]. The content of protein was estimated from the content of N, which was determined in a LECO CHNS-932 (LECO Corp., St. Joseph, MI, USA) elemental analyser, using a correction factor of 6.25 [24]. The ash content was measured by calcination of the sample in a muffle furnace at 575 °C for 6 h. Three replicates were used to determine the content of each component.

### 2.2. Isolation of the Milled-Wood Lignin from OTPs

The MWL was extracted from OTPs by aqueous dioxane according to the classical method, which allows for obtaining a lignin preparation with a largely unmodified chemical structure that is considered to reflect the structure of the native lignin [20]. About 60 g of extractive-free OTPs were ball-milled in a Restch PM-100 planetary mill (Retsch, Haan, Germany), using agate jar and balls at 400 rpm for 16 h (at intervals of 20 min of milling and 10 min of rest). Then, the finely milled sample was successively extracted with dioxane:water (90:10 *v*/*v*) for 18 h. The supernatant liquid, which contains the solubilized lignin, was separated from the solid via centrifugation. The extraction procedure was repeated two more times with a new dioxane:water solution, mixing the obtained supernatants each time. Then, the entire volume of the total collected supernatant liquids was brought to dryness via rotary evaporation at 40 °C. The crude MWL was subsequently purified by successive dissolution, precipitation, and centrifugation, using different solvent systems, following the procedure described elsewhere [25]. The MWL yields accounted for around 20% of the total OTP lignin content.

### 2.3. Gel Permeation Chromatography (GPC)

The OTP-MWL was first acetylated using acetic anhydride/pyridine (1:1, *v*/*v*) and subsequently dissolved in tetrahydrofuran (THF) for the GPC analyses. The analyses were carried out in a GPC system (Shimadzu Prominence-I LC-2030 3D) equipped with a 300 mm × 7.5 mm i.d., 5 μm, Lgel MIXED-D column and using a photodiode array (PDA) detector. The eluent used was THF (at 40 °C and 0.5 mL min^−1^ of flow rate). For data acquisition and processing, the LabSolution GPC version 5.82 software (Shimadzu, Kyoto, Japan) was used. A polystyrene standards kit (Agilent Technologies, Stockport, United Kingdom) was used to prepare the calibration curve, with an *M*_r_ range of 5.8 × 10^2^ to 3.24 × 10^6^ Da. 

### 2.4. Py-GC/MS Analysis of OTP-MWL

Pyrolysis was carried out using approximately 1 mg of OTP-MWL at a temperature of 500 °C (1 min) in a microfurnace pyrolizer (EGA/PY-3030D, Frontier Laboratories Ltd., Fukushima, Japan) coupled with an Agilent 7820A GC and an Agilent 5975 mass selective detector (Agilent Technologies, Inc., Santa Clara, CA, USA). The pyrolysis products were separated on a 30 m long × 0.25 mm internal diameter DB-1701 fused silica capillary column, with a film thickness of 0.25 µm. The oven temperature in the GC was heated from 50 °C (1 min) to 100 °C, at a rate of 30 °C min^−1^, and then to 280 °C (10 min) at 6 °C min^−1^. Helium was used as carrier gas (flow rate of 1 mL min^−1^). The identification of the compounds released after the pyrolysis was carried out by comparing their mass spectra with those of our own collection of standards, with those included in the Wiley and NIST mass spectra database, as well as with those described in the literature [26,27]. Molar peak areas were determined for each compound, normalizing the total areas and expressing the average of two measurements as percentages. 

### 2.5. 2D-NMR Analysis of OTP-MWL

For the 2D-NMR analyses, around 60 mg of OTP-MWL was dissolved in 0.7 mL of DMSO-*d*_6_. The heteronuclear single quantum coherence (HSQC) spectrum was acquired at 300 K on a 500 MHz Bruker AVANCE III instrument equipped with a 5 mm triple resonance inverse (TCI) cryogenically cooled probe. The 2D-HSQC-NMR experiments were acquired using the Bruker standard pulse sequence “hsqcetgpsisp2.2” (adiabatic-pulsed version), using the experimental conditions previously published [25]. The central solvent peak (δ_C_ 39.5; δ_H_ 2.49) was set as the internal reference. The different signals present in the NMR spectra were assigned via comparison with the literature [25,28,29].

The semiquantitative determination of the signals corresponding to the different lignin units and inter-unit linkages in the HSQC spectrum was accomplished using Bruker’s Topspin 3.5 processing software. The integration of signals was performed separately for the different regions of the spectrum. Hence, in the aromatic/unsaturated region of the spectrum, C_2_/H_2_ correlation signals were used to determine the relative abundances of the H-, G-, and S-lignin units. However, as the S_2,6_ and the H_2,6_ signals involve two pairs of proton carbons, their integration volumes were divided by two. On the other hand, in the oxygenated/aliphatic region of the spectrum, the relative abundances of sidechains involved in the different lignin linkages (**A**, **B**, **C**, **D**, and **F**) were estimated from the C_α_/H_α_ correlations (signals A_α_, B_α_, C_α_, D_α_, E_α_, and F_α_), except for the cinnamyl alcohol end-groups (substructure **I**), for which the C_γ_/H_γ_ correlations (signal I_γ_) were used. The relative abundance of cinnamaldehyde end-groups (**J**) was estimated through the integration of the J_7_ signal and comparison with the respective I_β_ signal.

## 3. Results and Discussion

### 3.1. Main Constituents of OTPs

The abundances of the main constituents of OTP residue, namely, the contents of acetone, methanol, and water-soluble extractives, Klason lignin, acid-soluble lignin, hemicelluloses, cellulose, proteins, and ash, are indicated in Table 1. The total content of extractives in OTPs amounted to 3.9%, including acetone (3.0%), methanol (0.6%), and water-soluble (0.3%) extractives, with values similar to those previously reported by other authors [7,8]. Likewise, the total lignin content (which is the sum of both the Klason lignin and the acid-soluble lignin contents) in OTPs amounted to 21.2%, which is also similar to previously reported values in the range of 20.0–22.2% [7,8,9]. OTPs also presented important carbohydrate contents, with hemicelluloses being 29.3% and cellulose 40.5%, which are consistent with the values reported in previous studies [8,9]. Finally, OTPs also contained small amounts of proteins (2.7%) and ashes (2.4%), similar to those previously reported [7,8,9].

To study the structural characteristics of the lignin in OTPs, the lignin must first be extracted. However, there is no universal protocol for the isolation of lignin from lignocellulosic materials, and due to the difficulties in isolating a representative lignin preparation from OTPs, we decided to isolate the so-called “milled-wood lignin” (MWL). The MWL preparation is largely chemically unmodified and is widely considered to represent native lignin in planta [21]. In addition, MWL preparations present good solubilities for subsequent 2D-NMR analyses. Ball milling is a necessary step to isolate MWL, and this may cause some structural modifications in the lignin. However, recent research has shown that ball milling has only a low impact on the lignin structure, although it may slightly reduce its molecular weight [30]. However, in this work, we used a low milling time (only 16 h) to reduce the possibility of significant structural changes. The OTP-MWL was then thoroughly analysed by GPC, Py-GC/MS, and 2D-NMR.

### 3.2. Molecular Weight of the OTP-MWL Determined by GPC

The weight-average (*M*_w_) and number-average (*M*_n_) molecular weights of the OTP-MWL, as well as the polydispersity (*M*_w_/*M*_n_), were determined from the GPC analysis. The data indicated that the OTP-MWL exhibited a *M*_w_ of 6385 g/mol and a *M*_n_ of 3774 g/mol, with a low polydispersity of 1.7, indicating the homogeneity of the isolated MWL. However, it is likely that the ball milling has somehow affected the molecular weights of the MWL [30]. In any case, these values are similar to those of other MWLs isolated from other hardwoods, such as eucalyptus, beech, and others [31,32,33]. However, it is known that the molecular weights of the MWLs can vary depending on several factors, such as the source of the lignin, the isolation method, and the analytical techniques used for determination. It is interesting to note that the molecular weight of the OTP-MWL was similar to that of the lignin derived from black liquors after soda/AQ cooking of OTP residue, which exhibited a *M*_w_/*M*_n_ of 6430/4890 (*M*_w_/*M*_n_ of 1.3). Since the lignin obtained from the black liquors after soda/AQ treatment has undergone significant degradation and structural alteration during the alkaline cooking process [11], this fact, therefore, indicates that the soda/AQ cooking process caused a repolymerization of the lignin degradation products.

### 3.3. Lignin Composition of the OTP-MWL by Py-GC/MS

The OTP-MWL was analysed using Py-GC/MS, an analytical technique that gives information about the lignin composition. Pyrolysis induces the thermal cleavage of the lignin into a set of different monomeric fragments, which maintain the original methoxylation pattern of the different lignin units (*p*-hydroxyphenyl, H; guaiacyl, G; and syringyl, S); therefore, it is a useful tool to estimate the H:G:S lignin composition [34,35]. Figure 1 shows the Py-GC/MS chromatogram of OTP-MWL, and Table 2 lists the identification and the relative abundances of the lignin-derived phenolic compounds produced upon pyrolysis.

The major compounds released during the pyrolysis of OTP-MWL were guaiacyl- and syringyl-type phenols, which were derived from the G- and S-lignin units, respectively. They included guaiacol (peak **2**), 4-methylguaiacol (peak **5**), 4-ethylguaiacol (peak **7**), 4-vinylguaiacol (peak **8**), eugenol (peak **9**), *cis*-isoeugenol (peak **11**), *trans*-isoeugenol (peak **12**), and vanillin (peak **14**), as well as the respective syringol (peak **10**), 4-methylsyringol (peak **13**), 4-ethylsyringol (peak **15**), 4-vinylsyringol (peak **18**), 4-allylsyringol (peak **19**), *cis*-4-propenylsyringol (peak **22**), *trans*-4-propenylsyringol (peak **23**), and syringaldehyde (peak **24**), among others. Phenolic compounds arising from H-lignin units, such as phenol (peak **1**), 3-methylphenol (peak **3**), 4-methylphenol (peak **4**), and 4-ethylphenol (peak **6**), could also be found, although in lower amounts. From the relative content of the different phenolic compounds released upon Py-GC/MS, it was possible to estimate the H:G:S composition of the OTP-MWL, which revealed that it was slightly enriched in G-lignin units (H:G:S of 1:57:42; S/G of 0.73), as indicated in Table 2. The predominance of G-lignin units in OTPs would make this lignocellulosic material more arduous and difficult to delignify via alkaline cooking due to the lower reactivity of G-lignin observed in alkaline systems compared to S-lignin [36]. From a structural point of view, and contrary to the S-units, G-units have the C5 position free to form additional condensed carbon–carbon linkages or ether inter-unit linkages, which will produce a highly condensed and branched lignin structure that will make them quite resistant to lignin depolymerization during alkaline cooking. It is clear, then, that the predominance of G-lignin units present in OTPs will imply lower delignification rates, as well as higher alkali consumption and, therefore, lower pulp yields [34]. It should be noted that the lignin composition observed in OTP-MWL was completely different from that of the lignin obtained from the black liquors during soda/AQ cooking of OTPs, which exhibited a strong predominance of S-lignin units and a depletion of G-lignin units, with a high S/G ratio of 4.6 [11]. This fact clearly indicated that the lignin from OTPs was largely chemically modified during the soda/AQ cooking process due to the preferential solubilization of S-lignin units in alkaline systems, as described previously for other lignocellulosic materials [37].

### 3.4. Lignin Units and Lignin Inter-Unit Linkages of the OTP-MWL by 2D-NMR

The OTP-MWL was also analysed using 2D-HSQC-NMR, which gives information not only about the composition of the lignin units, but also about the different lignin inter-unit linkages. The HSQC spectrum of the OTP-MWL is shown in Figure 2, and the different lignin units and substructures identified are also shown at the bottom. The HSQC spectrum was divided into two parts: the aliphatic/oxygenated region (δ_C_/δ_H_ 50–90/2.5–6.0), which provided information on the lignin inter-unit linkages, and the aromatic/unsaturated region (δ_C_/δ_H_ 98–155/5.8–7.8), which provided information on the composition of the H-, G-, and S-lignin units. The different lignin correlation signals observed in the HSQC spectrum of the OTP-MWL and their assignments are listed in Table 3.

The aromatic/unsaturated region of the spectrum showed signals from syringyl (**S**), guaiacyl (**G**), and *p*-hydroxyphenyl (**H**) lignin units. S-lignin units presented a strong signal for the C_2,6_/H_2,6_ correlations at δ_C_/δ_H_ 103.9/6.69, while G-lignin units presented different correlation signals for C_2_/H_2_, C_5_/H_5_, and C_6_/H_6_ (at δ_C_/δ_H_ 110.9/6.97, 114.8/6.93, and 118.8/6.77). Signals for the C_2,6_/H_2,6_ correlations of Cα-oxidized S-lignin units (**S′**) were present at δ_C_/δ_H_ 106.2/7.30 and 106.4/7.18. Finally, a small signal could also be detected for the C_2,6_/H_2,6_ correlations of H-lignin units at δ_C_/δ_H_ 127.7/7.20, indicating the low abundance of H-lignin units in OTP-MWL, as was also revealed by the Py-GC/MS analysis. Other signals in this part of the spectrum came from the unsaturated sidechains and aromatic units from cinnamyl alcohol (**I**) and cinnamaldehyde end-groups (**J**), as well as from spirodienones (**F**).

The correlation signals observed in the aliphatic/oxygenated part of the spectrum provided information regarding the different lignin inter-unit linkages. In this region of the spectrum, besides the signal of the methoxyl groups (at δ_C_/δ_H_ 55.5/3.72), the most predominant signals corresponded to β–*O*–4′ alkyl–aryl ether linkages (**A**). The C_α_/H_α_ and C_β_/H_β_ correlations of β–*O*–4′ alkyl–aryl ethers were slightly shifted and presented different signals depending on if they were linked to G- or S-units. Thus, the C_α_/H_α_ correlations were found at δ_C_/δ_H_ 70.9/4.73 for β–*O*–4′ substructures linked to G-units and at δ_C_/δ_H_ 71.8/4.83 for β–*O*–4′ substructures linked to S-lignin units. Similarly, the C_β_/H_β_ correlations were found at δ_C_/δ_H_ 83.5/4.28 for β–*O*–4′ substructures linked to G-units and at δ_C_/δ_H_ 85.8/4.10 for β–*O*–4′ substructures linked to S-lignin units. The C_γ_/H_γ_ correlations in β–*O*–4′ substructures were found at δ_C_/δ_H_ 59.7/3.38 and 3.69. Signals from other substructures, including phenylcoumarans (**B**), resinols (**C**), dibenzodioxocins (**D**), and spirodienones (**F**), as well as from cinnamyl alcohol end-groups, were also found in this area of the spectrum. Among them, the most intense signals were from β–5′ phenylcoumarans (**B**), with their C_α_/H_α_ and C_β_/H_β_ correlations being present at δ_C_/δ_H_ 86.7/5.45 and 53.0/3.45, and those of their C_γ_/H_γ_ correlations overlapping with other signals around δ_C_/δ_H_ 62.5/3.67. Intense signals for β–β′ resinols (**C**) were also present in this region of the spectrum, with their C_α_/H_α_, C_β_/H_β_ and the double-C_γ_/H_γ_ correlations at δ_C_/δ_H_ 84.8/4.65, 53.4/3.05, and 70.9/3.81 and 4.18, respectively. Signals for 5–5′ dibenzodioxocins (**D**) were also clearly found, with their C_α_/H_α_ and C_β_/H_β_ correlations being present at δ_C_/δ_H_ 83.1/4.81 and 85.2/3.84, respectively. Small signals from β–1′ spirodienones (**F**) could also be seen in the spectrum, with their characteristic signals for the C_α_/H_α_, C_α′_/H_α′_, C_β_/H_β_, and C_β′_/H_β′_ correlations being observed at δ_C_/δ_H_ 81.0/5.04, 83.5/4.71, 59.5/2.74, and 79.2/4.11. Another small signal in this part of the spectrum came from the C_γ_/H_γ_ correlations of the *p*-hydroxycinnamyl (**I**) end-groups at δ_C_/δ_H_ 61.3/4.08.

The relative abundances of the main lignin inter-unit linkages, lignin end-groups, and lignin aromatic units in OTP-MWL were estimated from the 2D-HSQC-NMR spectrum, and the data are displayed in Table 4. The data showed that OTP-MWL was predominantly constituted of G-lignin units (62% of all aromatic units), followed by S-lignin units (37%) and small amounts of H-lignin units (1%), with an S/G ratio of 0.60, in line with the information obtained via the Py-GC/MS analysis. The higher proportion of G-lignin units in OTP-MWL has a strong influence on the abundance of the inter-unit linkages observed in this lignin. Hence, and although β–*O*–4′ alkyl–aryl ethers were the most abundant linkages in OTP-MWL (representing up to 70% of the total linkages detected), important amounts of condensed linkages occurred in this lignin, including phenylcoumarans (15%) and resinols (9%), as well as minor amounts of dibenzodioxocins (3%) and spirodienones (3%). Cinnamyl alcohol and cinnamaldehyde end-groups represented up to 6% each, with respect to the total inter-unit linkages. The presence of dibenzodioxicins in OTP-MWL was noteworthy, since these condensed substructures are usually found in softwood lignin, where they act as branching points [38,39], but are hardly observed in the lignin from hardwoods; therefore, it is evident that the occurrence of significant amounts of dibenzodioxicins in OTP-MWL is due to the significant amounts of G-units found in this lignin.

It is important to note that the composition and structure of OTP-MWL completely differs from that of the lignin isolated from the black liquors during the soda/AQ cooking process [11]. The soda/AQ lignin was highly enriched with syringyl units (with a high S/G ratio of 4.6) and resinol structures, while other linkages (particularly β–*O*–4′ alkyl–aryl ethers and phenylcoumarans) were present in a much lower abundance [11]. It is evident that the lignin from soda/AQ cooking underwent a strong chemical modification during the alkaline treatment and was no longer representative of the native lignin in OTPs. 

As mentioned above, lignocellulosic materials with significant contents of G-lignin units (low S/G ratios) are more difficult to delignify via alkaline cooking and require larger amounts of alkali than lignocellulosic materials with higher S/G ratios [34,36]. Therefore, the enrichment with G-lignin units, as well as the significant amounts of condensed linkages (particularly phenylcoumarans and dibenzodioxocins) observed in the OTP-MWL are indicative that, under alkaline delignification, the lignin from OTPs would be less reactive than lignin from other hardwoods with lower contents of G-units (higher S/G ratios) and, consequently, with a lower abundance of condensed linkages (such as eucalyptus wood) [33,35]. Additionally, prunings from olive trees are another commonly available agricultural residue. Compared to OTPs, the lignin from olive prunings has a higher S/G ratio (S/G of 0.9) and, consequently, a higher content of β–*O*–4′ alkyl–aryl ether linkages (75%) and a lower content of condensed linkages [29]. As a result, the variations in the lignin composition between the two pruning residues indicate that OTPs may be slightly more resistant to alkaline treatments compared to the olive pruning residues.

## 4. Conclusions

The structural characteristic of the lignin from OTP residues, with a 21.2% lignin content, was thoroughly studied. For this, the MWL, a lignin preparation representative of the native lignin, was extracted from OTPs and thoroughly analysed using different analytical techniques. The analysis indicated that the OTP-MWL was enriched in G-lignin units, followed by S-lignin units and low amounts of H-units (H:G:S 1:62:37; S/G ratio 0.6). The main linkages were β–*O*–4′ alkyl–aryl ethers (70%), followed by phenylcoumarans (15%) and resinols (9%), with lower amounts of dibenzodioxocins (3%) and spirodienones (3%). The significant lignin content (21.2%) present in the OTP residues, along with the low S/G ratio and the relatively high content of condensed lignin structures, indicates a low reactivity of OTP lignin during alkaline delignification compared to other hardwoods with higher S/G ratios and lower contents of condensed linkages, such as eucalyptus wood. It should be noted that the lignin content and composition may slightly differ depending on various environmental and cultivation factors, including cultivar, crop location, harvesting age, or year of harvesting. To assess the variations in lignin composition among samples from different sources, additional orange tree samples should be analysed. Nonetheless, the information presented in this work marks the first time that the detailed lignin composition and structure of OTP residues have been disclosed, thereby paving the way for complete industrial utilization of these residues from a biorefinery standpoint. It is expected that the comprehensive understanding of the structure of the OTP lignin carried out here will help design effective deconstruction strategies, as well as the development of high-value-added lignin-based products from this interesting lignocellulosic agroindustrial waste material for biorefinery purposes.

## Figures and Tables

**Figure 1 polymers-15-01840-f001:**
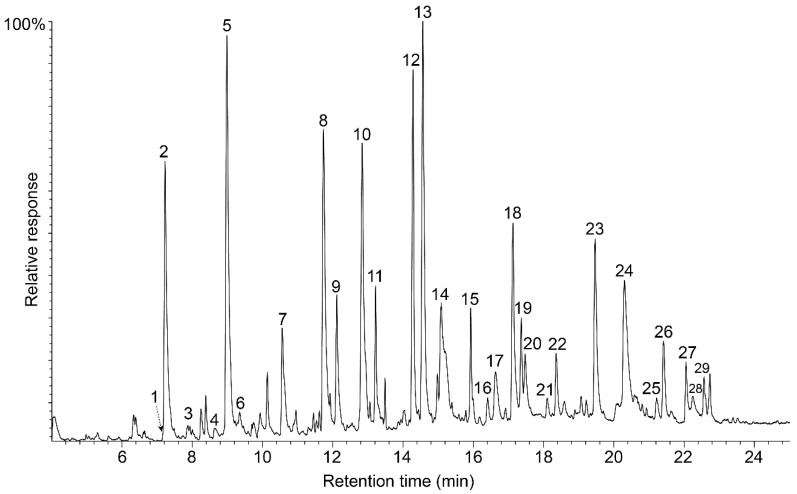
Py-CG/MS of the OTP-MWL. The identities and relative abundances of the released numbered compounds are listed in Table 2.

**Figure 2 polymers-15-01840-f002:**
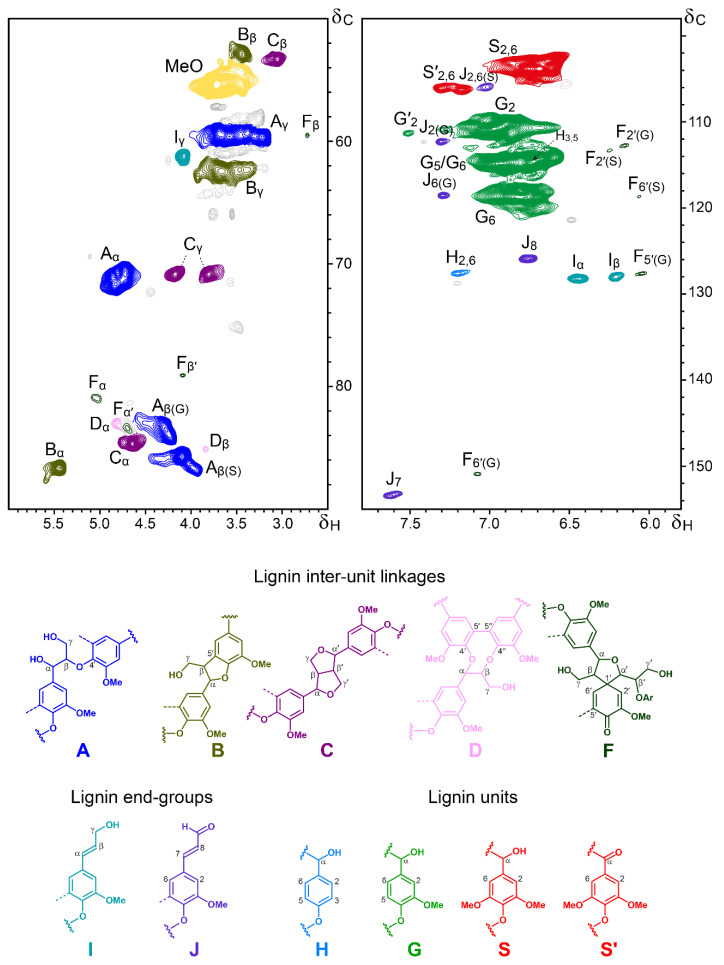
Sidechain (δ_C_/δ_H_ 50–90/2.5–6.0; left) and aromatic (δ_C_/δ_H_ 98–155/5.8–7.8; right) regions of the HSQC spectrum (in DMSO-*d*_6_) of the OTP-MWL. The lignin structures identified are shown below: **A**: β–*O*–4′ alkyl–aryl ethers; **B**: phenylcoumarans; **C**: resinols; **D**: dibenzodioxocins; **F**: spirodienones; **I**: *p*-hydroxycinnamyl alcohol end-groups; **J**: *p*-hydroxycinnamaldehyde end-groups; **H**: *p*-hydroxyphenyl units; **G**: guaiacyl units; **S**: syringyl units; **S′**: α-oxidized syringyl units.

**Table 1 polymers-15-01840-t001:** Abundance (in percentage) of the main constituents of OTP residue (dry basis).

	Abundance *^a^*
Acetone extractives	3.0 ± 0.3
Methanol extractives	0.6 ± 0.0
Water-soluble material	0.3 ± 0.0
Klason lignin *^b^*	19.0 ± 0.4
Acid-soluble lignin	2.2 ± 0.3
Hemicelluloses	29.3 ± 1.8
Cellulose	40.5 ± 0.3
Proteins	2.7 ± 0.1
Ash	2.4 ± 0.0

*^a^* Average of three replicates. *^b^* Corrected for proteins and ash.

**Table 2 polymers-15-01840-t002:** Identities and relative molar abundances (%) of the phenolic compounds released upon Py-GC/MS of the OTP-MWL. The retention time, mass spectrometric fragments, and origin of the released compounds are included.

Label	Compound	Ret. Time (Min)	MS Fragments	Origin	%
1	phenol	7.15	65/66/94	H	0.1
2	guaiacol	7.21	81/109/124	G	9.5
3	3-methylphenol	7.88	77/107/108	H	0.3
4	4-methylphenol	8.60	77/107/108	H	0.5
5	4-methylguaiacol	9.01	95/123/138	G	12.9
6	4-ethylphenol	9.35	77/107/122	H	0.4
7	4-ethylguaiacol	10.51	122/137/152	G	3.0
8	4-vinylguaiacol	11.70	107/135/150	G	9.0
9	eugenol	12.11	131/149/164	G	3.4
10	syringol	12.90	111/139/154	S	8.6
11	*cis*-isoeugenol	13.20	131/149/164	G	2.5
12	*trans*-isoeugenol	14.29	131/149/164	G	7.1
13	4-methylsyringol	14.65	125/153/168	S	9.0
14	vanillin	15.09	109/151/152	G	4.6
15	4-ethylsyringol	15.93	107/167/182	S	3.8
16	vanillic acid methyl ester	16.32	123/151/182	G	0.6
17	acetoguaiacone	16.64	123/151/166	G	1.8
18	4-vinylsyringol	17.12	137/165/180	S	4.2
19	4-allylsyringol	17.37	167/179/194	S	1.2
20	guaiacylacetone	17.47	122/137/180	G	1.4
21	propiovanillone	18.10	123/151/180	G	0.3
22	*cis*-4-propenylsyringol	18.37	167/179/194	S	1.0
23	*trans*-4-propenylsyringol	19.45	167/179/194	S	4.2
24	syringaldehyde	20.29	167/181/182	S	4.7
25	syringic acid methyl ester	21.15	123/181/212	S	0.8
26	acetosyringone	21.40	153/181/196	S	2.0
27	syringylacetone	22.05	123/167/210	S	1.3
28	*trans*-coniferaldehyde	22.24	135/147/178	G	0.8
29	propiosyringone	22.56	151/181/210	S	1.0
				%H=	1.3
				%G=	56.9
				%S=	41.8
				S/G=	0.73

H: *p*-hydroxyphenyl units; G: guaiacyl units; S: syringyl units. Underlined mass fragments indicate base peaks in their mass spectra.

**Table 3 polymers-15-01840-t003:** Assignments of ^13^C/^1^H correlation signals in the 2D-NMR (HSQC) spectrum of the OTP-MWL.

Label	δ_C_/δ_H_	Assignment
B_β_	53.0/3.45	C_β_/H_β_ in β–5′ phenylcoumarans (**B**)
C_β_	53.4/3.05	C_β_/H_β_ in β–β′ resinols (**C**)
-OCH_3_	55.5/3.72	C/H in methoxyls
A_γ_	59.7/3.38 and 3.69	C_γ_/H_γ_ in β–*O*–4′ alkyl–aryl ethers (**A**)
F_β_	59.5/2.74	C_β_/H_β_ in β–1′ spirodienones (**F**)
I_γ_	61.3/4.08	C_γ_/H_γ_ in cinnamyl alcohol end-groups (**I**)
B_γ_	62.5/3.67	C_γ_/H_γ_ in β–5′ phenylcoumarans (**B**)
A_α(G)_	70.9/4.73	C_α_/H_α_ in β–*O*–4′ alkyl–aryl ethers (**A**) linked to G-units
C_γ_	70.9/3.81 and 4.18	C_γ_/H_γ_ in β–β′ resinols (**C**)
A_α(S)_	71.8/4.83	C_α_/H_α_ in β–*O*–4′ alkyl–aryl ethers (**A**) linked to S-units
F_β′_	79.2/4.11	C_β′_/H_β′_ in β–1′ spirodienones (**F**)
F_α_	81.0/5.04	C_α_/H_α_ in β–1′ spirodienones (**F**)
D_α_	83.1/4.81	C_α_/H_α_ in 5-5′ dibenzodioxocins (**D**)
A_β(G)_	83.5/4.28	C_β_/H_β_ in β–*O*–4′ alkyl–aryl ethers (**A**) linked to G-units
F_α′_	83.5/4.71	C_α′_/H_α′_ in β–1′ spirodienones (**F**)
C_α_	84.8/4.65	C_α_/H_α_ in β–β′ resinols (**C**)
D_β_	85.2/3.84	C_β_/H_β_ in 5-5′ dibenzodioxocins (**D**)
A_β(S)_	85.8/4.10	C_β_/H_β_ in β–*O*–4′ alkyl–aryl ethers (**A**) linked to S-units
B_α_	86.7/5.45	C_α_/H_α_ in β–5′ phenylcoumarans (**B**)
S_2,6_	103.9/6.69	C_2_/H_2_ and C_6_/H_6_ in etherified syringyl units (**S**)
J_2,6(S)_	106.0/7.03	C_2_/H_2_ and C_6_/H_6_ in sinapaldehyde end-groups (**J**)
S′_2.6_	106.2/7.30 and 106.4/7.18	C_2_/H_2_ and C_6_/H_6_ in C_α_-oxidized syringyl units (**S′**)
G_2_	110.9/6.97	C_2_/H_2_ in guaiacyl units (**G**)
J_2(G)_	112.4/7.31	C_2_/H_2_ in coniferaldehyde end-groups C5-linked (**J**)
F_2′(G)_	112.8/6.16	C_2′_/H_2′_ in guaiacyl β–1′ spirodienones (**F**)
F_2′(S)_	113.4/6.25	C_2′_/H_2′_ in syringyl β–1′ spirodienones (**F**)
G_5_/G_6_	114.8/6.93	C_5_/H_5_ in guaiacyl units + C_6_/H_6_ in C5-linked guaiacyl units (**G**)
G_6_	118.8/6.77	C_6_/H_6_ in guaiacyl units (**G**)
J_6(G)_	118.7/7.30	C_6_/H_6_ in coniferaldehyde end-groups C5-linked (**J**)
F_6′(S)_	118.8/6.06	C_6′_/H_6′_ in syringyl β–1′ spirodienones (**F**)
J_8_	125.9/6.75	C_8_/H_8_ in cinnamaldehyde end-groups (**J**)
H_2,6_	127.7/7.20	C_2_/H_2_ and C_6_/H_6_ in *p*-hydroxyphenyl units (**H**)
F_5′(G)_	127.8/6.05	C_5′_/H_5′_ in guaiacyl β–1′ spirodienones (**F**)
I_β_	128.1/6.21	C_β_/H_β_ in cinnamyl alcohol end-groups (**I**)
I_α_	128.3/6.44	C_α_/H_α_ in cinnamyl alcohol end-groups (**I**)
F_6′(G)_	151.0/7.08	C_6′_/H_6′_ in guaiacyl β–1′ spirodienones (**F**)
J_7_	153.5/7.62	C_7_/H_7_ in cinnamaldehyde end-groups (**J**)

**Table 4 polymers-15-01840-t004:** Structural characteristics (inter-unit linkages, *p*-hydroxycinnamyl end-groups, H-, G-, and S-lignin units, and S/G ratios) from integration of the signals in the 2D-NMR (HSQC) of the MWL isolated from OTP residue.

Lignin Inter-Unit Linkages (%)	
β–*O*–4′ aryl ethers (**A**)	70
β–5′ phenylcoumarans (**B**)	15
β–β′ resinols (**C**)	9
5–5′ dibenzodioxocins (**D**)	3
β–1′ spirodienones (**F**)	3
**Lignin end-groups *^a^***	
Cinnamyl alcohol end-groups (**I**)	6
Cinnamaldehyde end-groups (**J**)	6
**Lignin aromatic units *^b^***	
H (%)	1
G (%)	62
S (%)	37
S/G ratio	0.60

*^a^* Expressed as a fraction of the total lignin inter-unit linkage types **A**-**F**. *^b^* Molar percentages (H + G + S = 100).

## Data Availability

Not applicable.

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
