# Peer review of "Structural Characterization of the Milled-Wood Lignin Isolated from Sweet Orange Tree (Citrus sinensis) Pruning Residue"

_polymers, 2023, doi:10.3390/polym15081840_

Round 1
Reviewer 1 Report
Attach file

Author Response
The manuscript ID Polymers-2196716 is entitled: “Structural characterization of the milled wood lignin isolated from sweet orange (Citrus sinensis) tree pruning residues”.
The authors described the importance and use of orange tree pruning residues of Citrus sinensis present 21.2% of lignin content.
For determination of the lignin in orange tree pruning was used Py-GC/MS and 2DNMR. The results indicated that the lignin was mainly composed by guaiacyl (G), followed by syringyl (S) and minor amounts of p-hydroxyphenyl (H) units (H:G:S composition of 1:62:37).
The main constituents of orange tree pruning of Citrus sinensis residues were cellulose and hemicellulose.
The authors described 29 major compounds identified of the pyrolysis analysis and they were classified in p-hydroxyphenyl units, guaiacyl units and syringyl units.
The lignin analyzed by 2D-HSQC-NMR was important to determine the lignin interunit linkages and the different lignin units. The HSQC analysis provided information on the lignin inter-unit linkages and the aromatic/unsaturated region.
The figures and tables are well discussed and are in good quality and importance.
In conclusion, the authors consider this subject as great development of high value added lignin-based products from lignocellulosic of waste material for biorefinery purposes.
The manuscript is suitable for publication in Polymers Journal.
Thanks for your comments and your support to our work. We appreciated it very much!
Minor corrections:
Page 1, line 1, Title: … Sweet Orange (Citrus sinensis) Tree … change by …
Sweet Orange (Citrus sinensis) Tree …
Corrected!
Page 1, line 29: … United States (4.3 M tonnes), Egypt (3,5 M tonnes)… check Egypt (3,5 M tonnes) or Egypt (3.5 M tonnes) ?
Corrected!
Page 3, line 100: … a muffle furnace at 575 ºC for 6 h … change by … a muffle furnace at 575 °C for 6 h …
Corrected!
Suggestion: check throughout the manuscript
Thanks for catching all these errors. They all were corrected!
Reviewer 2 Report
The article submitted for review presents consistent and precisely described results of the analysis of the structure of lignin extracted from milled orange tree pruning (OTP). I have no doubts about the quality of the results presented in this section.
However, the results concern only one sample (The orange tree pruning residues (…) were collected in July 2021 from a plantation in Seville (Spain)), therefore in my opinion it cannot be considered representative of all OTPs, especially since biological material chemical composition may vary seasonally. Have the Authors considered a such issue?
The second, debatable issue is the assumption that after OTP milling, the lignin retained their native structure. According to the knowledge available to me, the chemical structure of wood elements changes under the influence of mechanical forces, e.g. fragmentation (compare: Sommer, A., Staroszczyk, H., Sinkiewicz, I. & Bruździak, P. Preparation and Characterization of Films Based on Disintegrated Bacterial Cellulose and Montmorillonite. J Polym Environ 29, 1526–1541 (2021).). Even the Authors of manuscript suggest that the lignin molecular weight profile resembles that of the lignin obtained from the black liquors after soda/AQ cooking of OTP, p. 5, v. 189). Considering the above, I propose to supplement the research with non-destructive methods, e.g. NIR spectra before and after extraction/grinding, which will allow for assessing changes in the chemical structure (or confirm the preservation of the native structure of lignin after grinding).
Additionally, I have the following minor comments:
1) page 2, line 92: should be ash (not ashe)
2) p. 3, l. 95: what do the authors mean by "α-cellulose"? the concept has not been explained. Section 2.1 in materials and methods is long and complicated; In my opinion, it is worth supplementing this fragment with patterns or a graphic scheme of the procedure
3) Please at least briefly describe the crude lignin purification method (p. 3, l. 114, The crude MWL was purified following the procedure described elsewhere)
Author Response
The article submitted for review presents consistent and precisely described results of the analysis of the structure of lignin extracted from milled orange tree pruning (OTP). I have no doubts about the quality of the results presented in this section.
However, the results concern only one sample (The orange tree pruning residues (…) were collected in July 2021 from a plantation in Seville (Spain)), therefore in my opinion it cannot be considered representative of all OTPs, especially since biological material chemical composition may vary seasonally. Have the Authors considered a such issue?
The reviewer's observation is correct, as the lignin content and composition may slightly differ depending on various environmental and cultivation factors, including cultivar, crop location, harvesting age, or year of harvesting. To assess the variations in lignin composition among samples from different sources, additional orange tree samples should be analyzed. Nonetheless, the information presented in this work marks the first time that the detailed lignin composition and structure of orange tree pruning residues have been disclosed, thereby paving the way for complete industrial utilization of these residues from a biorefinery standpoint. A sentence explaining this has been included in the Conclusion section.
The second, debatable issue is the assumption that after OTP milling, the lignin retained their native structure. According to the knowledge available to me, the chemical structure of wood elements changes under the influence of mechanical forces, e.g. fragmentation (compare: Sommer, A., Staroszczyk, H., Sinkiewicz, I. & Bruździak, P. Preparation and Characterization of Films Based on Disintegrated Bacterial Cellulose and Montmorillonite. J Polym Environ 29, 1526–1541 (2021).). Even the Authors of manuscript suggest that the lignin molecular weight profile resembles that of the lignin obtained from the black liquors after soda/AQ cooking of OTP, p. 5, v. 189). Considering the above, I propose to supplement the research with non-destructive methods, e.g. NIR spectra before and after extraction/grinding, which will allow for assessing changes in the chemical structure (or confirm the preservation of the native structure of lignin after grinding).
There is not a universal protocol for the isolation of lignin from lignocellulosic materials and, due to the difficulties in isolating a representative lignin preparation from OTP, we decided to isolate the so-called “milled wood lignin” (MWL). The MWL preparation is largely chemically unmodified and is widely considered to represent native lignin in plants, as noted by different authors (Fujimoto et al., 2005; Rencoret et al., 2009). MWL is the method for lignin isolation with the lowest structural modifications, providing a lignin preparation representative of the native lignin and with good solubility for subsequent 2D-NMR analyses. Ball milling is necessary to isolate MWL, but this may cause some structural modifications in the lignin, as noted by this reviewer. However, the low milling time used in this work (only 16 h) reduces the likelihood of significant structural changes. Moreover, recent research indicates that ball milling mainly affects the high molar mass fraction of lignin by slightly reducing its molecular weight, but has only a low impact on the lignin structure (Zinovyev et al., 2018). A sentence has been added to Section 3.1 to clarify this point.
On the other hand, the reviewer suggests using NIR for assessing changes in the lignin structure during milling. However, NIR does not provide significant structural details for the lignin polymer. Nonetheless, we routinely use in our analyses a more powerful non-degradative method, such as 2D-NMR spectroscopy, for in situ lignin analysis (in the whole cell wall, without prior lignin isolation), and concluded that the main lignin structures as well as the S/G ratios matched those obtained from the isolated MWL, which would also indicate that MWL is still an appropriate and reliable lignin preparation representative of native lignin that can be used for comprehensive lignin structural studies (see our paper Rencoret et al., 2009).
Additionally, I have the following minor comments:
1) page 2, line 92: should be ash (not ashe)
Thanks for catching the error, it was corrected!
2) p. 3, l. 95: what do the authors mean by "α-cellulose"? the concept has not been explained. Section 2.1 in materials and methods is long and complicated; In my opinion, it is worth supplementing this fragment with patterns or a graphic scheme of the procedure
Well, α-cellulose is a concept commonly used in the cellulose (pulp and paper) industry and refers to non-degraded cellulose. We have now changed it in this work for the more appropriate “cellulose”.
3) Please at least briefly describe the crude lignin purification method (p. 3, l. 114, The crude MWL was purified following the procedure described elsewhere)
A short paragraph has been included to briefly describe the lignin purification process: “The crude MWL was subsequently purified by successive dissolution, precipitation and centrifugation using different solvent systems following the procedure described elsewhere”
Reviewer 3 Report
Thanks to the editor for inviting me to review the manuscript entitled “Structural Characterization of the Milled Wood Lignin Isolated from Sweet Orange (Citrus sinensis) Tree Pruning Residues.”
The authors investigated for the first time the structural characteristics of pristine lignin derived from alternative Orange tree pruning wastes. The results have high reference values for related research and application. Therefore, I recommend accepting it after major revision.
Comments:
1- The general structure is acceptable, the results seem good as well and the discussion is fine. However, language should be thoroughly revised as some of the sentences are confusing and some errors can be found. Check the whole document.
2- I recommend adding a schematic illustration of the preparation procedure of MWL from OTP to improve the impact of the paper.
3- The chemical composition of OTP and the molecular weight of its MWL derivative should be compared with recent literature reports on lignin obtained from different renewable sources.
4- The discussion of the results presented in section 3.2 needs a deep improvement in order to improve the quality of the manuscript.
5- It will be very interesting if the authors could add additional important structural characterizations including FTIR, SEM, and XRD measurements to improve the content and the impact of the paper.
6- The references part should be updated and some recent works in the field should be cited.
Author Response
Thanks to the editor for inviting me to review the manuscript entitled “Structural Characterization of the Milled Wood Lignin Isolated from Sweet Orange (Citrus sinensis) Tree Pruning Residues.”
The authors investigated for the first time the structural characteristics of pristine lignin derived from alternative Orange tree pruning wastes. The results have high reference values for related research and application. Therefore, I recommend accepting it after major revision.
Thanks for your comments and for the support to our work!
Comments:
1- The general structure is acceptable, the results seem good as well and the discussion is fine. However, language should be thoroughly revised as some of the sentences are confusing and some errors can be found. Check the whole document.
Thanks for the comment. The text has been carefully checked, and some errors were identified and corrected, while the language was improved wherever necessary.
2- I recommend adding a schematic illustration of the preparation procedure of MWL from OTP to improve the impact of the paper.
We appreciate the suggestion by this reviewer. However, we do not see the need to incorporate further illustrations for the MWL isolation protocol. The procedure has been established as a standard practice to isolate lignin from lignocellulosic materials for many years and has been extensively detailed and documented in numerous references.
3- The chemical composition of OTP and the molecular weight of its MWL derivative should be compared with recent literature reports on lignin obtained from different renewable sources.
The molecular weights have been compared with literature values for other hardwoods, and some additional references have been added.
4- The discussion of the results presented in section 3.2 needs a deep improvement in order to improve the quality of the manuscript.
The discussion in section 3.2 has been improved.
5- It will be very interesting if the authors could add additional important structural characterizations including FTIR, SEM, and XRD measurements to improve the content and the impact of the paper.
Thanks for the suggestion. However, in our opinion, the techniques used in this work, and more specifically 2D-NMR, are known to provide the most complete information on the structural characteristics of lignin (lignin units, and inter-unit linkages), which is the aim of this work.
6- The references part should be updated and some recent works in the field should be cited.
Thanks for the comment. The references have been updated where appropriate and more recent references have been included.
Round 2
Reviewer 3 Report
The authors corrected all the necessary issues. I believe that the manuscript can be published in its current form.